# BID: Broad Incremental for Android Malware Detection

## Abstract

With the rapid rise of mobile devices, the threat of malware targeting these platforms has escalated significantly. The fast-paced evolution of Android malware and new attack patterns frequently introduce substantial challenges for detection systems. Although many methods have achieved excellent results, they need to be retrained when faced with new attack modes or observation objects, and it is challenging to attain dynamic updates. To address this issue, we propose a novel Broad Incremental Detection (BID) method for real-time Android malware detection. Our method leverages incremental function to achieve dynamic adaptation to the growing variety of malware attacks while maintaining high computational efficiency, benefiting from its lightweight shallow network architecture. We also develop relational structures to capture complex relations and features of history attacks by fine-turning the network's weights unsupervised. Experimental results across three datasets demonstrate that BID achieves superior detection accuracy and computational efficiency compared to state-of-the-art approaches. Our work presents a robust, flexible, and lightweight framework for dynamic Android malware detection.

## 1 Introduction

With the widespread adoption of mobile devices, particularly smartphones, the Android operating system (OS) has emerged as a dominant force. Compared to its counterparts, such as iOS and Windows, Android enjoys a significantly larger global user base, holding a substantial share of the mobile device market. However, this proliferation of Android devices has escalated security threats Razgallah et al. (2021). Android has become the primary target for mobile malware, which can infiltrate devices through various means, including app downloads, malicious links, and network vulnerabilities. This exposes users' personal information, banking details, passwords, and more. Therefore, designing an effective Android malware detection system is an urgent necessity.

According to previous research, Android malware detection technology can be mainly categorized into three types: static detection Pan et al. (2020), dynamic detection García & DeCastro-Garcia (2021), and hybrid detection Hadiprakoso et al. (2020). Static detection involves analyzing suspicious code without running Android applications. In contrast, dynamic detection is based on analyzing Android applications by running the code. Hybrid detection combines both static and dynamic detection methods. However, as obfuscation technology advances and becomes more prevalent, traditional rule-based Mehtab et al. (2020) detection methods struggle to keep up with these rapidly evolving threats. Specifically, they often suffer from overfitting, decreased classification accuracy, and increased false positive rates when encountering new malware. Recently, deep learning Gopinath & Sethuraman (2023), Aslan & Yilmaz (2021), Shaukat et al. (2023) has been widely adopted for Android malware detection. These methods automatically extract features from many collected samples through reverse analysis, enhancing adaptability to new malware variants and improving detection accuracy. Although deep learning has certain advantages in malware detection, it has several limitations, e.g., longer training time, higher computational costs, and more extensive parameter tuningBensaoud et al. (2024). Moreover, with the continuous evolution of malware and attack techniques, retraining deep learning models to identify new malware becomes highly time-consuming and labour-intensive.

As an efficient alternative to deep neural networks, the broad learning system (BLS) Chen & Liu (2017), which is based on the random vector functional link neural network (RVFLNN) Pao et al. (1994), has attracted more attention due to its outstanding performance and shorter training time. BLS is a single-layer structural neural network, including feature nodes and enhancement nodes. In general, feature nodes are obtained from the original data, and enhancement nodes are mapped using a linear combination of feature nodes. Unlike stacking layers to improve accuracy, BLS expands in a broad direction. The output of the final weight is calculated by pseudo-inverse, resulting in short training time and not requiring high hardware conditions. Simultaneously, incorporating incremental learning into BLS allows for real-time parameter updates and system reconstruction as new malware samples emerge without retraining. This ensures that the system remains responsive and up-to-date, making it highly suitable for the dynamic and rapidly evolving landscape of Android malware detection.

Additionally, due to the typically large number of features involved in Android malware detection, feature selection is necessary to enhance model interpretability and prevent overfitting. However, BLS generates mapping features by randomly initializing connection weights. To overcome randomness, sparse autoencodersNg et al. (2011) are employed to fine-tune and select features by minimizing the loss function, which consists of reconstruction function and regularization, demonstrating good ability in extracting meaningful features. However, sparse autoencoders only consider data reconstruction while ignoring the relationships and structure between the data. To address this issue, we propose using a Sparse Relational Autoencoder (SRAE) to minimize the loss of its data features and the relationships among them.

To address the challenge of rapidly evolving malware patterns and to improve feature selection, we propose a unified framework Broad Incremental Detection (BID) for Android malware detection. Here, the main contributions of this paper are given as follows:

1) We are the *first* to employ an incremental function that enables the BID to dynamically adapt to new malware samples without retraining, ensuring both efficiency and real-time malware detection.

2) To capture the complex relationships and features of history attacks, we develop relational structures to fine-tune the network weights unsupervised.

3) Experiment results show that BID achieves significant improvements in performance and speed compared to machine learning and deep learning, benefiting from its lightweight network architecture.

## 2 Related Work and Background

### 2.1 Android Malware

Android malware, specifically refers to those malicious program codes that are crafted against the Android operating system with the aim of compromising the integrity, confidentiality, and availability of the device and its data. This type of malware comes in various forms and covers a wide range of types such as Trojans, ransomware, spyware and adware Alqahtani et al. (2019).

Malware refers to any type of malicious program code that can be installed automatically or stealthily on all types of devices without the user's explicit consent and performs its predefined malicious functions without the user being aware of it Agrawal & Trivedi (2019). Currently, a notable feature of Android malware is its ability to evade detection by traditional antivirus solutions Wu et al. (2021), and to achieve infiltration through advanced technical means such as hidden code and altered payloads. To ensure persistence on infected devices, these malware may also employ sophisticated methods such as masquerading as a system application or installing a rootkit, making removal more difficult.

A major challenge of Android malware detection is its dynamic and evolving nature. Malware creators continue to develop new variants and use advanced techniques to evade existing detection systems. This adaptability allows malware to modify behavioral patterns and conceal code, making it difficult for static and signature-based detection mechanisms to cope Wang et al. (2020).

Overall, the continuous evolution of Android malware presents significant challenges to traditional detection mechanisms. As new variants emerge and adapt, there is an increasing need for more robust and intelligent detection methods that can respond to these changes effectively.

## 2.2 EXISTING METHODS

**Rule-based Detection**: Traditional malware detection methods primarily rely on rule-based approaches that utilize predefined rules or features to identify malware. Early techniques focused on signature-based detection Sihag et al. (2020), which detects malware by comparing file features against a database of known malware. Behaviour-based detection Tanana (2020) identifies malicious activities by monitoring the runtime behaviour of programs using established rules. Additionally, permission-based detection Şahin et al. (2023) analyzes the permissions requested by Android applications upon installation to identify potential malware. While rule-based methods can be effective in specific scenarios, they face limitations, including poor adaptability to new malware and vulnerability to variant attacks.

**DL-based Detection**:Deep learning (DL) methods have gained widespread application in malware detection in recent years, leveraging large volumes of training data and complex models to capture latent patterns and characteristics. For instance, Dong et al. (2024) and Wang et al. (2020) employ convolutional neural networks (CNN) to classify malware, achieving significant performance improvements by training on raw byte streams. García et al. (2023) enhances the detection capabilities of deep learning models for new malware samples through transfer learning. While DL methods often outperform traditional rule-based approaches in accuracy and robustness, they also encounter challenges, such as high data requirements and substantial computational resource consumption.

In contrast, we propose the *first* BL-based malware detection approach. Benefiting from its lightweight shallow network, the broad incremental function enables dynamic adaptation to evolving attack patterns while maintaining high computational efficiency and low resource consumption.

## 2.3 BROAD LEARNING SYSTEM

Inspired by the Random Vector Functional Link Neural Network (RVFLNN), BLS differs by not directly connecting its input and output layers. BLS constructs its hidden layer using $n$ groups of feature nodes and $m$ groups of enhancement nodes. Feature nodes and enhancement nodes are obtained via random mapping functions.

Given input data $\mathbf{X} = \{\mathbf{x}_1, \mathbf{x}_2, \ldots, \mathbf{x}_n\}$ and labels $\mathbf{Y} = \{\mathbf{y}_1, \mathbf{y}_2, \ldots, \mathbf{y}_n\}$, the feature mapping nodes are computed as:

$$\mathbf{Z}_i = \phi(\mathbf{X}\mathbf{W}_{fi} + \boldsymbol{\beta}_{fi}), \quad i = 1, 2, \ldots, n, \tag{1}$$

where $\mathbf{W}_{fi}$ and $\boldsymbol{\beta}_{fi}$ are randomly sampled, and $\phi$ is the activation function Chen & Liu (2017). The feature mapping layer is denoted as $\mathbf{Z}^n = [\mathbf{Z}_1, \mathbf{Z}_2, \ldots, \mathbf{Z}_n]$. Enhancement nodes are calculated by:

$$\mathbf{E}_j = \zeta(\mathbf{Z}^n\mathbf{W}_{ej} + \boldsymbol{\beta}_{ej}), \quad j = 1, 2, \ldots, m, \tag{2}$$

where $\mathbf{W}_{ej}$ and $\boldsymbol{\beta}_{ej}$ are randomly generated, and $\zeta$ is typically chosen as the *tansig* function. The enhancement layer is denoted as $\mathbf{E}^m = [\mathbf{E}_1, \mathbf{E}_2, \ldots, \mathbf{E}_m]$. The hidden layer is a fusion of feature and enhancement nodes: $\mathbf{H} = [\mathbf{Z}^n \mid \mathbf{E}^m]$. The output is obtained via:

$$\mathbf{Y} = \mathbf{H}\mathbf{W}, \tag{3}$$

where $\mathbf{W}$ is the output weight matrix. To solve for $\mathbf{W}$, we minimize:

$$\mathbf{W} = \underset{\mathbf{W}}{\arg\min} : \|\mathbf{H}\mathbf{W} - \mathbf{Y}\|_2^2 + \lambda\|\mathbf{W}\|_2^2, \tag{4}$$

with $\lambda$ preventing overfitting. The solution is:

$$\mathbf{W} = \mathbf{H}^+\mathbf{Y} = \lim_{\lambda \to 0}(\lambda I + \mathbf{H}^\top\mathbf{H})^{-1}\mathbf{H}^\top\mathbf{Y}. \tag{5}$$

## 3 PROBLEM STATEMENT

Let $f_\theta : \mathbf{X} \to \mathbf{Y}$ be a learning model that maps features of Android applications (such as API calls, permission requests, and behavioral patterns) from an input feature space $\mathbf{X}$ to an output label space

$\mathbf{Y}$, where $\mathbf{Y}$ represents the category of the application (e.g., malware or benign). By optimizing the model parameters $\theta$ over a training dataset $(\mathbf{X}^{(train)}, \mathbf{Y}^{(train)})$, we aim to ensure that $f_\theta$ achieves high classification accuracy on a test dataset $(\mathbf{X}^{(test)}, \mathbf{Y}^{(test)})$.

However, due to the rapid evolution of Android malware, new data $\mathbf{X}_{new}$ may contain previously unseen features, which makes it challenging for the model to maintain high performance. This results in a potential decline in detection accuracy when encountering these novel data. Addressing this issue is crucial for building a robust, real-time malware detection system capable of handling the dynamic nature of Android malware.

# 4 PROPOSED METHOD

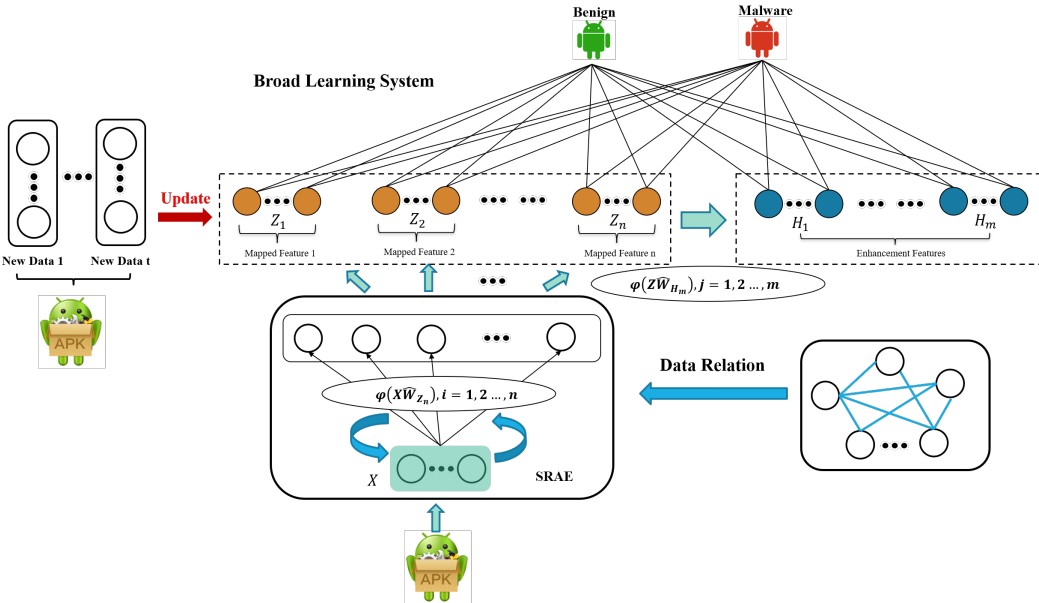

Figure 1: The workflow of the our method.

Figure 1 overviews our method. First, the detection process begins by collecting data from Android applications, such as behaviour patterns, permissions, and network activity. The relationships and structures between these data points are analyzed, and essential features are extracted to reduce complexity. Second, the extracted features are fed into the our framework, which classifies the app as either malicious or benign. Finally, when new variants of Android malware appear, they are also processed through the BID. One of the advantages of this approach is that BID does not need to be retrained when new data is added, allowing the system to classify new malware quickly without extra training steps. This ensures that malware can be detected quickly and effectively, even as it evolves.

In details, we initially define the input as $\mathbf{X} = \{x_1, x_2, \ldots, x_n\}$ and the label matrix as $\mathbf{Y} = \{y_1, y_2, \ldots, y_n\}$. Let $\mathbf{Z} \in \mathbb{R}^{n \times k}$ be the randomly generated feature matrix computed by Equation (1), where $n$ is the sample size and $k$ is the number of transformed features.

Since BID generates the mapping features by randomly initializing the connecting weights, in order to overcome the randomness, a sparse relational autoencoder is adopted to more effectively capture data relationships and give a sparse representation. As we can see, the random features $\mathbf{Z}$ are generated as equation $\mathbf{Z} = \mathbf{XW}$, where $\mathbf{W}$ is randomly initialized. Thus, the SRAE loss function is formulated as:

$$\min_{\tilde{\mathbf{W}}_{\mathbf{Z}^n}} (1-\alpha)\|\mathbf{Z}\tilde{\mathbf{W}}_{\mathbf{Z}^n} - \mathbf{X}\|_2^2 + \alpha\|\tau_t(\mathbf{Z}\mathbf{Z}^\top)\tilde{\mathbf{W}}_{\mathbf{Z}^n} - \tau_t(\mathbf{X}\mathbf{X}^\top)\|_2^2 + \lambda\|\tilde{\mathbf{W}}_{\mathbf{Z}^n}\|_2^2 \qquad (6)$$

Here, $\alpha$ balances the data reconstruction and relationship reconstruction losses, while $\lambda$ is the regularization weight. The gradient with respect to $\tilde{\mathbf{W}}$ is given by:

$$\nabla_{\tilde{\mathbf{W}}_{\mathbf{Z}^n}} = 2(1 - \alpha)(\mathbf{Z}^\top(\mathbf{Z}\tilde{\mathbf{W}}_{\mathbf{Z}^n} - \mathbf{X})) + 2\alpha(\tau_t(\mathbf{ZZ}^\top)\tilde{\mathbf{W}}_{\mathbf{Z}^n} - \tau_t(\mathbf{XX}^\top)) + 2\lambda\tilde{\mathbf{W}}_{\mathbf{Z}^n} \quad (7)$$

After determining $\tilde{\mathbf{W}}_{\mathbf{Z}^n}$, the mapping features are redefined as:

$$\mathbf{Z}_i = \xi_i(\mathbf{X}\tilde{\mathbf{W}}_{zi}), \quad i = 1, 2, \ldots, n \quad (8)$$

where $\tilde{\mathbf{W}}_{zi}$ are weights from $\tilde{\mathbf{W}}$ and $\xi_i(\cdot)$ is a nonlinear function, yielding $\mathbf{Z}^n = [\mathbf{Z}_1, \mathbf{Z}_2, \ldots, \mathbf{Z}_n]$. This steprefines the feature mapping process, enhancing the efficiency and effectiveness of the model.

Similarly, the refined enhancement node can be obtained through $\tilde{\mathbf{W}}_{\mathbf{E}^m}$, which is optimized by equation (9).

$$\min_{\tilde{\mathbf{W}}_{\mathbf{E}^m}} (1 - \alpha)\|\mathbf{F}\tilde{\mathbf{W}}_{\mathbf{E}^m} - \mathbf{E}\|_2^2 + \alpha\|\tau_t(\mathbf{FF}^\top)\tilde{\mathbf{W}}_{\mathbf{E}^m} - \tau_t(\mathbf{EE}^\top)\|_2^2 + \lambda\|\tilde{\mathbf{W}}_{\mathbf{E}^m}\|_2^2, \quad (9)$$

where the transformed features are denoted by $\mathbf{F} = \mathbf{E}^m W_{\mathbf{H}^m} \in \mathbb{R}^{N \times k_1}$ with $W_{\mathbf{H}^m}$ being randomly initialized.

Finally, the combined mapping and transformed feature nodes are given by $\mathbf{H} = [\mathbf{Z}^n | \mathbf{E}^m\tilde{\mathbf{W}}_{\mathbf{E}^m}]$, leading to the final weight:

$$\mathbf{W}^+ = (\lambda\mathbf{I} + \mathbf{H}^\top\mathbf{H})^{-1}\mathbf{H}^\top\mathbf{Y}. \quad (10)$$

**Incremental Learning:** In BLS, the incremental approach is based on calculating the pseudo-inverse of the partitioned matrix. It estimates the Moore-Penrose generalized inverse by incorporating a small positive value into the diagonal of $\mathbf{HH}^\top$, in accordance with the principles of ridge regression. Therefore, we can continue to modify our solutions by modifying $\mathbf{W}^+$. Let $\mathbf{A}_n^m$ represent the nodes of the initial network. The corresponding increment nodes for the new samples $\mathbf{x}$ can be expressed as follows:

$$\mathbf{H}_x = [\mathbf{Z}_\mathbf{x} \mid \mathbf{E}_\mathbf{x}]. \quad (11)$$

After that, we can combine the new and previous samples as,

$$\mathbf{H}^+ = \left[ \begin{array}{c} \mathbf{H}_n^m \\ \mathbf{H}_x \end{array} \right] \quad (12)$$

Specifically, $\mathbf{H}_N$ can represent data from a new malware sample or a new observation for the same sample in malware detection. We then update $\mathbf{W}^+$ by calculating the pseudo-inverse of the partitioned matrix. The algorithm for updating the associated pseudoinverse can be derived as follows:

$$(^x\mathbf{H}_n^m)^+ = \left[(\mathbf{H}_n^m)^+ - \mathbf{BD}^\top \mid \mathbf{B}\right], \quad (13)$$

$$\mathbf{B}^\top = \begin{cases} \mathbf{C}^+ & \text{if } \mathbf{C} \neq 0 \\ (1 + \mathbf{D}^\top\mathbf{D})^{-1} (\mathbf{H}_n^m)^+ \mathbf{D} & \text{if } \mathbf{C} = 0 \end{cases} \quad (14)$$

where $\mathbf{D}^\top = \mathbf{H}_\mathbf{x}\mathbf{H}_n^{m+}$ and $\mathbf{C}^+ = \mathbf{H}_\mathbf{x}^\top - \mathbf{D}^\top\mathbf{H}_n^m$. Finally, the dynamic updated weight is formulated as,

$$^x\mathbf{W}_n^m = \mathbf{W}_n^m + \left(\mathbf{Y}_\mathbf{x}^\top - \mathbf{H}_\mathbf{x}^\top\mathbf{W}_n^m\right)\mathbf{B} \quad (15)$$

where $\mathbf{Y}_\mathbf{x}$ is the label of new data $\mathbf{x}$. This incremental learning approach optimizes computation by only calculating the necessary pseudoinverse, making it ideal for handling new incoming input data, such as a new malware application or a new observation.

## 5 EXPERIMENT

In this section, the experiments are conducted to verify the performance of our model. Compared with several machine learning and deep learning method. All the experiments in this paper are carried out on four NVIDIA GeForce RTX 3090 GPUs.

### 5.1 DATASETS

1) The Tezpur University Android Malware Dataset (TUANDROMD) is publicly available at https://www.kaggle.com/datasets/joebeachcapital/tuandromd. For the experiments, we use both permission-based and API-based features of this dataset. Its features include 214 permissions and 27 unique API calls extracted from Android applications. The dataset contains 1000 benign samples from Google Play and 24,553 malware samples representing 71 distinct malware families.

2) The CIC-InvesAndMal-2019 dataset (CIC-2019) is publicly available at https://www.unb.ca/cic/datasets/invesandmal2019.html. For the experiments, we use the static analysis part of this dataset. Its features include 8115 permissions and intent behaviors extracted from the manifest.xml file of the APK file. The dataset contains 1187 benign samples and 407 malware samples. In addition to the basic binary classification benign and malware, malware is further categorized into the following five categories: a) adware; b) ransomware; c) scareware; d) SMS d) PremiumSMS.

3) The CCCS-CIC-AndMal-2020 dataset (CIC-2020) is publicly available at https://www.unb.ca/cic/datasets/andmal2020.html. The static analysis portion of the dataset contains 162,181 benign and 195,624 malware samples. The static analysis portion of the dataset contains 162,181 benign samples and 195,624 malware samples with 9,502 features related to permissions, intent, activity, broadcast receivers and providers, services, system characteristics, and metadata. Fourteen malware categories are covered, including adware, backdoors, file infectors, unclassified, potentially unwanted programs (PUAs), ransomware, riskware, scareware, Trojans, banking Trojans, droppers, SMS Trojans, spyware, and zero-day attackware.

### 5.2 BASELINES

We compare our proposed approach with the following baseline models. SVM Singh et al. (2022) is a classic classifier that finds a hyperplane to separate benign and malware classes by maximizing the margin. Bayesian Anggraini et al. (2023) is a probabilistic model that assumes feature independence to calculate the likelihood of each class. DeepAMD Brown et al. (2024) is a deep learning model designed specifically for Android malware detection, using multiple layers to extract high-level features from APK files. BiGRU Maniriho et al. (2023) is a Bidirectional Gated Recurrent Unit network that processes sequence data forward and backward to capture context from API calls. RNN-LSTM Al-Aql & Al-Shammari (2024) uses Long Short-Term Memory units to capture long-term dependencies in sequential data, making it practical for tasks like analyzing system call traces.

### 5.3 SETTINGS

Experimental dataset setup: we extracted the static dataset of CIC-2019, and 1/40 of the static dataset of CIC-2020, and the training set is set to 0.7. For BID with incremental learning added, we set the ratio of the training set, test set and incremental set to be 5:3:2.

To verify the effectiveness of BID, we selected three state-of-the-art deep learning methods: Deep-AMD, BiGRU and RNN-LSTM and two machine learning methods, SVM and Naive Bayesian, for comparison. For BiGRU, we set the number of GRUs to 8 and the dropout rate to 0.6; For Deep-AMD and RNN-LSTM, we set the number of hidden nodes in the middle layer to 10. All models use the same epochs (50) and batch size (64) to ensure fairness. For SVM, we used a nonlinear kernel function (RBF kernel). For the multi-categorization problem, a One-vs-One strategy is used to automatically train a binary classification model for every two categories between them, and a voting mechanism is used in the prediction phase to obtain the classification results. In addition, we use a polynomial Bayesian model, which is particularly suitable for discrete data and large-scale datasets and can show good results, especially when the feature dimensions are high or the number of classes is large.

## 5.4 RESULT

**Contrast experiment:** For the binary classification tasks presented in Table 1, the BID model consistently achieves the highest accuracy, precision, recall, and F1 score across different datasets. Specifically, in the TUANDROMD dataset, the model without increment reaches 99.48% in all metrics, demonstrating its robustness and efficiency with a relatively low time cost of 3.24 seconds. In comparison, other models like BiGRU and RNN-LSTM exhibit strong performance but with higher time costs, particularly in the CIC 2019 and CIC 2020 binary classification tasks, where BID still maintains its superiority, achieving similar top-tier results while minimizing computational overhead.

The BID model performs well for multiclass classification tasks, as shown in Table 2. In particular, the CIC 2019 multiclass dataset achieves an accuracy of 95.20% while maintaining the lowest time cost of 10.73 seconds. These experiments highlight the strong performance of BID across both binary and multiclass classification tasks, underscoring its versatility and suitability for a wide range of classification scenarios. The model's capacity to achieve high accuracy while maintaining short training time makes it well-suited for detecting Android malware.

Table 1: Performance Comparison on Binary Classification Datasets

| Model | Accuracy (%) | Precision (%) | Recall (%) | F1-Score (%) | Time (s) |
|-------|-------------|---------------|------------|--------------|----------|
| **TUANDROMD** | | | | | |
| SVM | 98.05 | 99.11 | 97.94 | 98.77 | 0.0798 |
| Bayesian | 94.10 | 95.50 | 99.50 | 96.31 | 0.0054 |
| DeepAMD | 98.06 | 98.15 | 98.06 | 98.79 | 17.389 |
| BiGRU | 97.91 | 98.02 | 97.91 | 98.35 | 33.001 |
| RNN-LSTM | 97.98 | 99.15 | 97.98 | 98.73 | 26.068 |
| BID | **99.48** | **99.48** | **99.48** | **99.48** | 3.24 |
| **CIC 2019** | | | | | |
| SVM | 88.28 | 88.91 | 88.28 | 87.33 | 1.66 |
| Bayesian | 89.95 | 89.94 | 89.95 | 85.52 | 0.02 |
| DeepAMD | 94.64 | 94.60 | 94.63 | 94.62 | 24.99 |
| BiGRU | 94.63 | 94.58 | 94.63 | 94.57 | 37.22 |
| RNN-LSTM | 94.79 | 94.75 | 94.79 | 94.73 | 19.02 |
| BID | **95.82** | **95.78** | **95.82** | **95.79** | 3.30 |
| **CIC 2020** | | | | | |
| SVM | 83.83 | 83.81 | 83.81 | 83.81 | 93.59 |
| Bayesian | 83.32 | 83.32 | 83.32 | 83.28 | 0.14 |
| DeepAMD | 92.05 | 92.16 | 92.06 | **94.07** | 94.27 |
| BiGRU | 91.23 | 91.76 | 91.23 | 91.25 | 185.90 |
| RNN-LSTM | 92.80 | **93.15** | 92.80 | 92.81 | 90.21 |
| BID | **92.99** | 93.11 | **92.99** | 93.00 | 88.79 |

**Increment experiment:** We divided each dataset into a training set, test set, and incremental set in a 5:3:2 ratio for the incremental experiments. The incremental dataset was sourced from the training set of the previous experiments. This setup simulates a real-world scenario where new malware samples become available over time, and the model needs to adapt without retraining from scratch.

In our incremental experiments, we observed improvements across all performance metrics after applying incremental learning, as presented in Table 3. All metrics are improved in all datasets. The consistent enhancements indicate that the BLS framework effectively leverages incremental data to enhance its malware detection capabilities. The model adapts to evolving malware patterns by incorporating incremental data, which is crucial for maintaining robust security measures in dynamic environments.

Moreover, we found that the total time for the training dataset (50%) and incremental dataset (20%) in the incremental experiment was less than the time required to train directly on the entire origi-

Table 2: Performance Comparison on CIC 2019 and CIC 2020 Multiclass Classification

| Model | Accuracy (%) | Precision (%) | Recall (%) | F1-Score (%) | Time (s) |
|-------|-------------|---------------|------------|--------------|----------|
| **CIC 2019** | | | | | |
| SVM | 84.31 | 85.70 | 84.30 | 81.00 | 2.11 |
| Bayesian | 76.98 | 76.41 | 76.98 | 69.84 | 0.02 |
| DeepAMD | 93.22 | 93.09 | 93.22 | 93.07 | 27.86 |
| BiGRU | 92.59 | 92.49 | 92.59 | 92.46 | 22.18 |
| RNN-LSTM | 92.43 | 92.33 | 92.43 | 92.32 | 40.20 |
| BID | **95.20** | **95.16** | **95.19** | **95.02** | 10.73 |
| **CIC 2020** | | | | | |
| SVM | 67.32 | 55.27 | 67.32 | 58.04 | 106.88 |
| Bayesian | 61.69 | 56.86 | 61.69 | 57.09 | 0.11 |
| DeepAMD | 82.28 | 77.83 | 82.28 | 79.13 | 88.42 |
| BiGRU | 84.85 | 82.27 | 84.85 | 82.71 | 191.01 |
| RNN-LSTM | 84.52 | 81.36 | 84.52 | 82.25 | 92.13 |
| BID | **85.79** | **84.72** | **85.79** | **84.38** | 83.00 |

nal training dataset (70%). This result highlights the computational efficiency of our incremental learning approach, as it reduces the overall training time while still enhancing performance. This efficiency is particularly beneficial for real-time malware detection systems, where timely updates are essential.

Table 3: Comparison of Experimental Results Before and After Data Increment

| Stage | Accuracy (%) | Precision (%) | Recall (%) | F1-Score (%) | Time (s) |
|-------|-------------|---------------|------------|--------------|----------|
| **TUANDROMD (Binary Classification)** | | | | | |
| Before | 98.58 | 98.58 | 98.58 | 98.58 | 1.44 |
| After | **99.33** | **99.33** | **99.33** | **99.33** | 0.70 |
| **CIC 2019 (Binary Classification)** | | | | | |
| Before | 93.93 | 93.87 | 93.93 | 93.86 | 2.39 |
| After | **95.82** | **95.78** | **95.82** | **95.79** | 0.31 |
| **CIC 2019 (Multiclass Classification)** | | | | | |
| Before | 92.25 | 92.86 | 92.26 | 92.36 | 3.73 |
| After | **94.35** | **94.48** | **94.35** | **94.25** | 0.31 |
| **CIC 2020 (Binary Classification)** | | | | | |
| Before | 92.13 | 92.21 | 92.13 | 92.14 | 16.70 |
| After | **92.95** | **93.09** | **92.95** | **92.26** | 0.45 |
| **CIC 2020 (Multiclass Classification)** | | | | | |
| Before | 85.15 | 83.91 | 85.15 | 84.05 | 14.33 |
| After | **85.75** | **84.64** | **85.75** | **84.38** | 0.51 |

## 6 CONCLUSION

In this paper, we introduced a novel framework (BID) for Android malware detection that utilizes an incremental learning approach to dynamically adapt to new malware variants without retraining. Our approach effectively balances detection accuracy and computational efficiency, benefiting from its lightweight and flexible network architecture. Our method enhances feature selection and improves detection capabilities by integrating relational structures to capture complex patterns from past malware attacks. Experimental results across multiple datasets demonstrate that our approach

outperforms existing methods, offering a robust and efficient solution for real-time Android malware detection.

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
