# OpenReview forum: "BID: Broad Incremental for Android Malware Detection"
_ICLR.cc/2025/Conference — Submitted to ICLR 2025_

### Official Review · Reviewer_JKeh · 2024-10-27

**Soundness:** 2
**Presentation:** 2
**Contribution:** 2
**Rating:** 1
**Confidence:** 5

**Summary:**

The paper presents a Broad Incremental Detection (BID) framework to address Android malware detection.
It proposes an incremental learning mechanism using the Broad Learning System (BLS) for
real-time malware detection without retraining, alongside Sparse Relational Autoencoders (SRAE)
for better feature selection. Experimental results are reported across three datasets,
suggesting that BID achieves superior performance and computational efficiency over existing models.

While this paper is timely and attempts to address a practical challenge in cybersecurity domain, there are several critical issues that need
to be addressed. In particular, the system is not clearly explained, the datasets are non-standard and limited,
and the results are not compared with state-of-the-art malware detection methods.
Important aspects, such as catastrophic forgetting and real-world applicability, are not addressed.
The experimental setup lacks depth with missing implementation details, no statistical analysis, and
an unclear discussion of results. See the detailed breakdown below.

**Strengths:**

1. Incremental Learning Integration: The application of incremental learning to Android malware detection is interesting and
        offers a practical way to reduce retraining costs.

2. Computational Efficiency: The paper shows that BID requires relatively less runtime compared to prior work, making it appear suitable for real-time scenarios.

3. Use of Public Datasets: The experiments use public datasets.

**Weaknesses:**

1. Clarity of the system

The reviewer had a hard time understanding how the proposed system works.
Furthermore, the use of BLS in an incremental learning framework is not particularly new,
especially in the malware domain (see related references 1-3, 12, 13).
The reviewer is unable to identify the unique contribution of the paper.

The reviewer suggests a step-by-step breakdown of the BID framework’s architecture and functioning.
Detailed explanation of how incremental learning is applied in the model,
beyond the BLS component would strengthen the paper.
In addition, references to similar works in malware
detection (e.g., specific papers on BLS in malware)
would enhance the comparison and clarify unique contributions of the paper.

2. Incremental learning (IL)

a. How is the proposed system better than other IL approaches?
b. How does IL compare with continual learning (CL) approaches for malware detection?
c. Why is IL needed instead of CL?

3. Catastrophic Forgetting for IL

The paper did not evaluate its system against catastrophic forgetting, a major drawback of incremental learning.

To ensure robustness, the reviewer recommends adding experiments to evaluate the
system against catastrophic forgetting phenomena,
a fundamental challenge in IL. The reviewer suggests the paper to follow this paper [14].


4. Lack of Comparison with State-of-the-Art Methods

The paper fails to compare its approach with widely adopted state-of-the-art
Android malware detection techniques (see related references 4-9).
This omission limits the relevance and impact of the reported findings.
Without such comparisons, it is difficult to determine whether the BID framework truly outperforms best practices.

The reviewer strongly suggests the paper to evaluate the system against these benchmarks to
evaluate the strengths of the proposed system.



5. Dataset

The most widely used Android malware dataset is Drebin [7], but the paper does not evaluate its approach with this dataset.
The datasets used (i.e., TUANDROMD, CIC-2019, and CIC-2020) are not widely adopted in the malware research space and are not considered benchmark datasets.

The paper uses only 1/40 of the CIC-2020 dataset, which is an insufficient sample size. It is unclear why the dataset was reduced for evaluation.

The explanation of the CIC-2020 dataset is inaccurate; the original paper mentions that the dataset contains 200K benign and 200K malware samples [10], but this paper reports different numbers.

The CIC-2019 dataset is unavailable at the given URL.

The datasets used are too small compared to standard practices in malware research.
Moreover, the ratio of benign to malware apps is typically 90:10 to reflect practical Android malware distributions [6],
but this ratio is not followed in the paper.

The reviewer suggests the paper to evaluate their system with i. Drebin dataset [7], and ii. use AndrooZoo [11] repository to collect a larger dataset following the best practices and evaluate their system against the larger dataset.

6. Experimental setting

Although the BID model shows some performance improvements, the experimental setup lacks depth. There are no details about the implementation or configuration of the proposed system. The reviewer suggests including hyperparameter choices, training configuration, and feature selection specifics.
These should be included to improve reproducibility.

The paper does not provide statistical significance analysis, error ranges, or detailed hyperparameter optimization procedures, making it difficult to trust the reported results. The reviewer suggests including standard deviation or confidence intervals on performance metrics.

The incremental dataset split is not well justified and does not seem to reflect the real-world dynamic evolution of malware.

7. Results

The reviewer finds the interpretation of the results unclear. There is no analysis or discussion of the results to provide insights into their significance.

The reviewer suggests the paper to include insights into
what the results imply about the system’s performance,
efficiency, and limitations.

8. No Discussion of Limitations and Real-world Impact:

The paper does not discuss the limitations of the proposed framework, such as challenges in real-world deployment. It is unclear how relevant this work is to practical Android malware detection systems.


Related references:

[1] Yuan, Wei, et al. "A lightweight on-device detection method for android malware." IEEE transactions on systems, man, and cybernetics: systems, 2019.

[2] Vasan, Danish, Mohammad Hammoudeh, and Mamoun Alazab. "Broad learning: A GPU-free image-based malware classification." Applied Soft Computing, 2024.

[3] Liu, Licheng, et al. "Self-paced broad learning system." IEEE Transactions on Cybernetics, 2022.

[4] McLaughlin, Niall, et al. "Deep android malware detection." CODASPY. 2017.

[5] Yuan, Zhenlong, et al. "Droid-sec: deep learning in android malware detection." SIGCOMM. 2014.

[6] Xu, Ke, et al. "Droidevolver: Self-evolving android malware detection system." EuroS&P 2019.

[7] Arp, Daniel, et al. "Drebin: Effective and explainable detection of android malware in your pocket." NDSS 2014.

[8] Mariconti, Enrico, et al. "Mamadroid: Detecting android malware by building markov chains of behavioral models." NDSS 2017.

[9] Renjith, G., and S. Aji. "On-device resilient Android malware detection using incremental learning." Procedia Computer Science 215 (2022): 929-936.

[10] Rahali, Abir, et al. "Didroid: Android malware classification and characterization using deep image learning." Proceedings of the 2020
10th International Conference on Communication and Network Security. 2020.

[11] Allix, Kevin, et al. "Androzoo: Collecting millions of android apps for the research community." Proceedings of the 13th international conference on mining software repositories. 2016.

[12] Vasan, Danish, Mohammad Hammoudeh, and Mamoun Alazab. "Broad learning: A GPU-free image-based malware classification." Applied Soft Computing 154 (2024): 111401.

[13] Zhang, Yibin, Guan Gui, and Shiwen Mao. "A lightweight malware traffic classification method based on a broad learning architecture." IEEE Internet of Things Journal (2023).

[14] Díaz-Rodríguez, Natalia, et al. "Don't Forget, There is More than Forgetting: new Metrics for Continual Learning." Continual Learning Workshop at NeurIPS 2018.

**Questions:**

1. How is the proposed system better than other Incremental Learning (IL) approaches?
2. How does IL compare with Continual Learning (CL) approaches for malware detection?
3. Why is IL preferred over CL in this context?
4. Why did the paper not include comparisons with widely adopted state-of-the-art Android malware detection techniques?
5. Why was the Drebin dataset not used for evaluation, given its widespread use in Android malware research?
6. Why was only 1/40 of the CIC-2020 dataset used, and how does this reduced size impact the evaluation?
7. Why does the paper report different numbers for the CIC-2020 dataset than the original source?
8. How does the incremental dataset split align with real-world malware evolution?
9. What strategies can be adapted to prevent catastrophic forgetting in the incremental learning framework for android malware detection?
10. What real-world deployment challenges do the authors foresee, and how does the system address them?

---

### Official Review · Reviewer_dhFa · 2024-10-27

**Soundness:** 2
**Presentation:** 2
**Contribution:** 2
**Rating:** 3
**Confidence:** 5

**Summary:**

The paper introduces the Broad Incremental Detection (BID) method for real-time Android malware detection. The main contributions of this work are (i) the ability to detect new malware samples without the need for model retraining, achieved using the Moore-Penrose pseudoinverse; (ii) the capability to capture relationships and historical attack features using a Sparse Relational Autoencoder (SRAE); and (iii) a lightweight architecture, inherent to the design of BID, which enables efficient deployment. Experimental results demonstrate that BID outperforms state-of-the-art methods in malware detection.

**Strengths:**

The paper combines different techniques to achieve what is claimed as a contribution. The manuscript is clear and well-written, presenting a good problem formalization.

**Weaknesses:**

* The paper proposes a combination of techniques (BLS + SRAE). However, the contribution appears to lie in applying these techniques to malware detection. The use of BLS for malware detection has been studied in [1], [2], [3]. Similarly, sparse autoencoders for malware detection were analyzed in [4]. The authors should consider including some of these works as a baseline for further comparison.

* The authors' second claim also appears biased: _"To capture the complex relationships and features of historical attacks, we develop relational structures to fine-tune unsupervised network weights."_ However, the authors rely on a Sparse Relational Autoencoder (SRAE), which is a well-established technique.

* The authors also claim that their approach is robust. However, labeled data is still required for retraining. Therefore, comparing it to traditional deep learning (DL) approaches, which can identify data patterns unsupervised, may be unfair. For instance, DL models can detect zero-day malware, whereas the proposed approach would still require labels. This limitation could be addressed by analyzing unseen samples during inference (without using the incremental technique) to observe how the proposal reacts and then comparing this behavior with traditional DL methods.

* In Equation (6), the parameter $\tau_t$ seems to be used as a threshold to balance the relationship between terms. However, this parameter is neither explained nor explored in the text. The authors should explain the $\tau_t$ parameter, including its purpose and how it is set, and conduct experiments to determine its optimal value. Similar experiments can be applied to $\alpha$ and $\lambda$ .

* A better explanation of the data used needs to be included. The authors should specify which samples are used for training and which are used for evaluation. It needs to be clarified which features are identified as the most important.

* In the results section, the authors state: _"demonstrating its robustness and efficiency with a relatively low time cost of 3.24 seconds." How is robustness evaluated in this experiment?_ What does "cost" refer to here (assumed to be time)? Are the authors referring to training or inference time? The authors should provide specific definitions and metrics for robustness and efficiency and clearly state whether the reported times refer to training or inference.

* The results in Table 1 also need further exploration. While the KPI on the TUANDROMD dataset shows good results, the time (presumably training time) appears to be up to four orders of magnitude higher, yet improves accuracy by only ~<1.5% compared to a traditional SVM. Additionally, more explanation is needed on the CIC 2020 dataset, particularly regarding why the proposed method seems to miss specific samples.

* The critique in Table 2 is similar regarding time. However, the authors need to specify the classified classes to allow the reproduction of the experiments.

* Table 3 is challenging to interpret, and more information about the experiments is necessary. What do the authors mean by "before" and "after" increment? I assume that "before" refers to the same data as in Tables 1 and 2, but the results differ. Please explain the experimental setup for Table 3, including definitions of "before" and "after" increments. Moreover, explain how these results differ from those in Tables 1 and 2.


[1] Vasan, Danish, Mohammad Hammoudeh, and Mamoun Alazab. "Broad learning: A GPU-free image-based malware classification." Applied Soft Computing 154 (2024): 111401.

[2] Wang, Shanshan, et al. "Deep and broad learning based detection of android malware via network traffic." 2018 IEEE/ACM 26th International Symposium on Quality of Service (IWQoS). IEEE, 2018.

[3] Liu, Guangyuan, Jiting Zhou, and Xiaoyu Ma. "Classification and Sharing Method of Malware Based on Threat Intelligence." 2020 IEEE 4th Information Technology, Networking, Electronic and Automation Control Conference (ITNEC). Vol. 1. IEEE, 2020.

[4] Zhu, Hui-Juan, et al. "A hybrid deep network framework for android malware detection." IEEE Transactions on Knowledge and Data Engineering 34.12 (2021): 5558-5570.

**Questions:**

* The authors mentioned in the introduction that _“BLS is a single-layer structural neural network, including feature and enhancement nodes.”_  It would be helpful if the authors could reflect the feature nodes in Figure 1.

* The authors should explain the method for selecting samples from each dataset. This will ensure that samples from one dataset do not overlap with those from others.

Minors comments:

* Several missing spaces were found: Lines [52, 69, 123, 213, 227]

---

### Official Review · Reviewer_zz1F · 2024-10-29

**Soundness:** 2
**Presentation:** 3
**Contribution:** 2
**Rating:** 3
**Confidence:** 4

**Summary:**

This paper proposal a Broad Incremental Detection (BID) method for real-time Android malware detection. This method leverages incremental function to achieve dynamic adaptation to the growing variety of malware attacks while maintaining high computational efficiency, benefiting from its lightweight shallow network architecture. The authors develop relational structures to capture complex relations and features of history attacks by fine-turning the network’s weights unsupervised. Experimental results across three datasets demonstrate that BID achieves good detection accuracy and computational efficiency compared to state-of-the-art approaches.

**Strengths:**

This paper addresses the important research problem of detecting Android malware, with a particular focus on the challenging task of identifying newly evolved malware. To achieve this, this work introduces an incremental function that allows the BID to dynamically adapt to emerging malware samples. The proposed approach is evaluated on three distinct datasets and benchmarked against three deep learning models and two machine learning models. The results highlight the good performance of the proposed method.

**Weaknesses:**

The primary concern with this work lies in the evaluation setup, which does not fully support the claim of detecting newly evolved malware. Although the work claims to identify new malware variants, the evaluation does not substantiate this. Specifically, in the "Increment experiment" section on Page 7, it is stated, "We divided each dataset into a training set, test set, and incremental set in a 5:3:2 ratio for the incremental experiments. The incremental dataset was sourced from the training set of the previous experiments." This setup implies that the testing and training datasets are drawn from the same datasets. To more robustly support the claim, the testing dataset should ideally include new malware families not represented in the training dataset, but this distinction is absent in the current experimental design.

Additionally, this work focuses on manually selected features, such as permissions and API calls, which malware authors can potentially evade. Many existing studies have instead analyzed malware code logic as a more robust detection method. Could you clarify the motivation for analyzing manually selected features rather than code logic analysis?

**Questions:**

1. Explain the experimental design of the "Increment experiment" part. Whether the testing datasets include new malware families not present in the training dataset? And how did you make sure of this?

2. Explain the motivation for analyzing manually selected features rather than code logic analysis.

---

### Official Review · Reviewer_mFRE · 2024-11-02

**Soundness:** 2
**Presentation:** 2
**Contribution:** 1
**Rating:** 3
**Confidence:** 4

**Summary:**

The  authors propose a Broad Incremental Detection (BID) method. This method leverages incremental function to achieve dynamic adaptation to the growing variety of malware attacks while maintaining high computational efficiency.

**Strengths:**

1. The topic of make ANDROID malware detection model evaluation is important.

2. The paper is well-organized.

**Weaknesses:**

The novelty of the paper is limited, as there have been papers in 2021 that utilized BLS for malware detection.

[1]A Lightweight On-device  Detection  Method  for  Android Malware, IEEE TSMC, 2021

**Questions:**

1.There have been papers in 2021 [1] that utilized BLS for malware detection. The authors need to at least present the distinctions in methodology/theory/application compared to [1].

[1]A Lightweight On-device  Detection  Method  for  Android Malware, IEEE TSMC, 2021

2.In the experiments, the authors should consider more realistic scenarios. For example, the authors could evaluate: 1) scenarios where data increases with years. This involves partitioning the dataset based on time periods and performing incremental learning with data from different years. 2) scenarios where data increases due to the emergence of new malicious families.

3.In the experimental section, the authors need to demonstrate the superiority of incremental learning. Compared to retraining models, what are the advantages of BLS's incremental training in terms of time and accuracy?

4.Authors mention 'developing relational structures to capture complex relations and features of historical attacks by fine-tuning the network's weights unsupervised,' but I don’t see specific analysis and related conclusions.

---

### Meta-Review · Area_Chair_skKt · 2024-12-20

**Metareview:**

The paper under review proposes a Broad Incremental Detection (BID) method for real-time Android malware detection. However, after careful consideration of the four reviewer comments, it is clear that the manuscript is not suitable for publication at this stage.

Reviewers unanimously pointed out the limited novelty of the paper. As noted by Reviewer mFRE, there have been papers in 2021 that utilized similar techniques (BLS) for malware detection. The authors failed to clearly present the distinctions in methodology, theory, or application compared to previous work. This lack of novelty significantly undermines the paper's contribution to the field. Reviewers zz1F, dhFa, and JKeh also expressed concerns about the paper's contribution. The combination of techniques proposed (BLS + SRAE) has been studied before, and the authors did not convincingly show how their application to malware detection is unique. The authors did not provide any responses in the rebuttal phase.

**Additional Comments On Reviewer Discussion:**

Multiple reviewers criticized the experimental setup. Reviewer zz1F noted that the evaluation does not fully support the claim of detecting newly evolved malware. The testing and training datasets appear to be drawn from the same datasets, which is not ideal for validating the claim of detecting new malware variants. Reviewer JKeh emphasized that the paper does not compare its approach with widely adopted state-of-the-art Android malware detection techniques. Without such comparisons, it is impossible to determine the superiority of the proposed BID framework.

---

### Decision · Program_Chairs · 2025-01-22

Reject